# LAT1-Targeted Alpha Therapy Using ^211^At-AAMT for Bone and Soft Tissue Sarcomas

**DOI:** 10.3390/ijms26178599

**Published:** 2025-09-04

**Authors:** Haruna Takami, Yoshinori Imura, Hidetatsu Outani, Sho Nakai, Akitomo Inoue, Yuki Kotani, Seiji Okada, Kazuko Kaneda-Nakashima

**Affiliations:** 1Department of Orthopaedic Surgery, Graduate School of Medicine, Osaka University, Osaka 565-0871, Japan; haruna.t3121@gmail.com (H.T.);; 2Radiation Biological Chemistry, MS-CORE, FRC, Graduate School of Science, Osaka University, Osaka 560-0043, Japan; kkaneda@irs.osaka-u.ac.jp; 3Institute for Radiation Sciences, Osaka University, Osaka 565-0871, Japan

**Keywords:** Astatine-211, L-type amino acid transporter 1, targeted alpha therapy, osteosarcoma, soft tissue sarcoma

## Abstract

Malignant bone and soft tissue tumors are often resistant to conventional treatment, and treatment options for unresectable and metastatic cases are limited. L-type amino acid transporter 1 (LAT1) is overexpressed in several malignancies, including sarcomas, making it an attractive target for targeted alpha therapy. In this study, we investigated the therapeutic efficacy of LAT1-targeted alpha therapy using a novel modified 3-astatin-211 Astato-α-methyl-L-tyrosine (^211^At-AAMT) for bone and soft tissue sarcomas. LAT1 expression and the specificity of LAT1-mediated uptake of ^211^At-AAMT were evaluated in bone and soft tissue sarcoma cell lines. Antiproliferative effects were assessed using cell viability and colony formation assays. DNA damage was assessed using immunostaining with phosphorylated histone γH2AX. In vivo efficacy of ^211^At-AAMT, determined using xenograft mouse models, was compared with that of doxorubicin. LAT1 was highly expressed in all cell lines, especially MP-CCS-SY and MG-63 cells. ^211^At-AAMT uptake was LAT1-dependent and significant in all cell lines. It inhibited cell proliferation in a dose-dependent manner, comparable to that of doxorubicin. In xenograft models, a single administration of ^211^At-AAMT significantly inhibited tumor growth without systemic toxicity, whereas doxorubicin caused weight loss. Histopathological analysis showed reduced cell density, inhibited proliferation, and extensive DNA damage in tumors treated with ^211^At-AAMT, whereas LAT1 expression was maintained in residual tumor tissues. LAT1-targeted alpha therapy with ^211^At-AAMT demonstrated antitumor efficacy comparable to that of first-line chemotherapy for osteosarcoma and soft tissue sarcoma. Sustained LAT1 expression suggests the potential for repeated or combination treatments, highlighting its promise as a novel therapy for advanced, treatment-resistant sarcomas.

## 1. Introduction

Sarcomas represent a rare category of cancers primarily comprising bone and soft tissue malignancies, accounting for approximately 1% of all diagnosed malignancies [1]. The standard treatment for resectable sarcomas is surgical resection with adequate margins, with or without adjuvant chemotherapy or radiotherapy [2]. However, for unresectable cases, effective local therapeutic options are limited to conventional photon radiotherapy, which often yields suboptimal results owing to the inherent radioresistance of sarcomas [3]. Heavy-ion radiotherapy has emerged as a revolutionary approach for treating unresectable sarcomas, as it enables the delivery of high-dose irradiation to the tumor while minimizing exposure to the surrounding normal tissues [4]. This modality has demonstrated promising outcomes in terms of local control [5]. However, heavy-ion radiotherapy is not indicated for advanced cases with metastases in Japan [6,7], and, currently, no effective radiotherapy options are available for metastatic sarcomas.

Targeted alpha therapy (TAT) has recently gained attention as a promising strategy for treating various malignancies, particularly those resistant to traditional therapies [8]. Alpha particles possess high linear energy transfer properties that induce double-strand breaks in the DNA, resulting in potent cytotoxic effects [9]. In addition, these particles have an extremely short range (50–100 μm) within living tissues, maximizing their tumor-killing effect while reducing damage to the surrounding normal tissues [9]. TAT facilitates systemic administration of alpha-emitting radionuclides as an internal radiation therapy, allowing simultaneous targeting of multiple foci, including metastatic sites [10]. These characteristics make TAT particularly well-suited for treating disseminated or multi-focal disease. Among the radionuclides being studied for TAT, astatin-211 (^211^At) has received considerable attention owing to its adequate half-life of 7.2 h, high energy alpha emission, and feasibility of production using accelerator-based technology [11]. One of the promising molecular targets for nuclear medicine therapy is the L-type amino acid transporter 1 (LAT1), a transmembrane protein responsible for the uptake of essential amino acids [12]. LAT1 is highly overexpressed in several malignancies, including malignant bone and soft tissue sarcomas, but is minimally expressed in normal tissues [13,14].

We have previously developed LAT1-targeted radiopharmaceuticals using 3-[^211^At] Astato-α-methyl-L-tyrosine (^211^At-AAMT-OH-L), a novel alpha-emitting therapeutic agent specifically designed to target LAT1-expressing tumors [15]. Compared with other TAT candidates, ^211^At-AAMT-OH-L has been reported to exert rapid effects that may reduce the burden on patients and rapid elimination that may reduce side effects, along with high affinity to LAT1 and strong cytotoxicity of alpha rays [15]. More recently, we further optimized this compound by introducing structural modifications, including methyl, ethyl, and propyl substitutions, to improve its stability and tumor retention and decrease nonspecific accumulation [16]. Among the variants developed, the methyl-substituted derivative ^211^At-AAMT-O-Me-L (^211^At-AAMT) exhibited the highest tumor uptake and retention with minimal off-target accumulation, making it the most promising candidate for clinical application [16]. The chemical structures of ^211^At-AAMT-OH-L and ^211^At-AAMT-O-Me-L are shown in Appendix A. Preclinical xenograft studies using a human pancreatic cancer cell line have confirmed its efficacy in inhibiting tumor growth with limited accumulation in normal tissue [16].

In this study, we aimed to evaluate the antitumor effects of this novel TAT (^211^At-AAMT-O-Me-L) in malignant bone and soft tissue tumor cells compared with first-line chemotherapeutic agents currently used for bone and soft tissue sarcomas.

## 2. Results

### 2.1. Expression of LAT1 and Specificity of LAT1-Mediated Uptake of ^211^At-AAMT in Bone and Soft Tissue Sarcoma Cell Lines

To investigate the expression of the LAT1 protein in four osteosarcoma and four soft tissue sarcoma cell lines, we performed Western blot analysis under both reducing (DTT(+)) and non-reducing (DTT(−)) conditions. LAT1 formed a heterodimer with CD98, and this linkage was disrupted by DTT treatment. HT-1080 cells served as the positive control. The LAT1-CD98 heterodimer was observed at 150 kDa, and the LAT1 and CD98 monomers were detected at 39 kDa and 75 to 120 kDa, respectively (Figure 1A). Densitometric quantification of LAT1-CD98 heterodimer bands, normalized to β-actin, revealed variable expression levels across cell lines (Appendix A). Although all examined cell lines demonstrated relatively high LAT1 expression, MP-CCS-SY and MG-63 cells exhibited particularly high expression levels. Next, to assess the specificity of LAT1-mediated uptake of ^211^At-AAMT in these cell lines, we measured ^211^At-AAMT uptake in the presence or absence of the dual LAT1/LAT2 inhibitor 2-aminobicyclo [2.2.1] heptane-2-carboxylic acid (BCH) and the selective LAT1 inhibitor nanvuranlat. Normal human dermal fibroblasts (NHDFs) were used as normal tissue controls. The MP-CCS-SY, Kitra-SRS, and MG-63 cell lines demonstrated high specificity for the LAT1-mediated uptake of ^211^At-AAMT (Figure 1B). Based on these findings, MP-CCS-SY and MG-63 cell lines were selected for further evaluation of the antitumor efficacy of ^211^At-AAMT.

### 2.2. ^211^At-AAMT Suppressed Cell Growth and Colony Formation in MP-CCS-SY and MG-63 Cells

To evaluate the growth-inhibitory effects of ^211^At-AAMT, we treated MP-CCS-SY and MG-63 cells with 0, 0.1, 1, 10, and 100 kBq and compared the antiproliferative effects of ^211^At-AAMT with those of doxorubicin, a first-line chemotherapy agent for bone and soft tissue sarcomas. ^211^At-AAMT significantly inhibited cell growth in a dose-dependent manner (Figure 2A), with the antiproliferative effects of 0.1–1 kBq ^211^At-AAMT comparable to those of 10 nM doxorubicin. The IC50 values at 48 h were comparable between the two cell lines. For doxorubicin, the IC50 was 31.0 nM (95% CI, 21.3–45.0; n = 3) in MP-CCS-SY cells and 70.7 nM (95% CI, 44.2–113.0; n = 3) in MG-63 cells. For ^211^At-AAMT, the IC50 was 100.8 kBq (95% CI, 37.5–271.0; n = 3) in MP-CCS-SY cells and 88.8 kBq (95% CI, 16.5–478.8; n = 3) in MG-63 cells (Appendix A). Furthermore, we assessed the effect of ^211^At-AAMT on colony formation. Treatment with ^211^At-AAMT for 48 h suppressed colony formation in both cell lines in a dose-dependent manner. In MP-CCS-SY cells, the inhibitory effect of 3 kBq ^211^At-AAMT appeared comparable to that of 10 nM doxorubicin (Figure 2B).

Moreover, ^211^At-AAMT treatment induced DNA damage owing to the formation of DNA double-strand breaks (DSBs) (Figure 3), as evidenced by a significant increase in the total fluorescence intensity of γH2AX in the 100 kBq group relative to the 0 kBq group (Appendix A), and by a dose-dependent increase in the mean number of γH2AX foci per cell (Appendix A). These results suggest that a relatively low dose of ^211^At-AAMT exerts significant antiproliferative effects on the bone and soft tissue sarcoma cells via DNA DSBs, comparable to those of doxorubicin.

### 2.3. ^211^At-AAMT Suppressed the Tumor Growth in the Xenograft Model

To investigate the in vivo antitumor effects of ^211^At-AAMT, we established xenograft models using human sarcoma and osteosarcoma cell lines with different levels of LAT1 expression. We evaluated LAT1 expression in tumor tissues and found that MP-CCS-SY cells exhibited high expression (score: 4), whereas VA-ES-BJ and SYO-1 cells showed low expression (score: 1), in accordance with the in vitro findings (Figure 1A and Figure 4A). As MG-63 cells failed to form tumors in nude mice, we used the 143B cell line to assess the antitumor effects of ^211^At-AAMT in vivo.

In MP-CCS-SY xenograft models, treatment with doxorubicin at 3 mg/kg did not significantly inhibit tumor growth compared with the vehicle control (Figure 4B). However, doxorubicin at 6 mg/kg significantly inhibited tumor growth as well as resulted in a 25% reduction in the average body weight by day 14, indicating severe off-target toxicity (Figure 4B and Appendix A). By contrast, a single administration of ^211^At-AAMT at a dose of 1 MBq significantly inhibited tumor growth (*p* = 0.008) without any noticeable reduction in the average body weight. Similarly, in 143B xenograft models, a single administration of ^211^At-AAMT resulted in remarkable tumor growth suppression, and consistent results were observed in terms of tumor weight (*p* = 0.001) (Figure 4B and Appendix A).

### 2.4. Histopathological Evaluation After ^211^At-AAMT Irradiation

HE staining revealed decreased cell density in the ^211^At-AAMT-treated group compared with that in the vehicle control group (Figure 5A,B). The proportion of Ki-67-positive cells was significantly lower in the ^211^At-AAMT-treated group than in the control (MP-CCS-SY: *p* < 0.001; 143B: *p* = 0.001) and doxorubicin-treated groups (MP-CCS-SY: *p* = 0.003; 143B: *p* = 0.003), indicating a potent antiproliferative effect of ^211^At-AAMT (Figure 5A,B). Furthermore, histological analysis at 24 h post-treatment revealed a significantly higher proportion of γH2AX-positive cells in the ^211^At-AAMT group than in the doxorubicin group (MP-CCS-SY: *p* = 0.01; 143B: *p* < 0.001), suggesting that ^211^At-AAMT exerts its antitumor effect through DNA damage, consistent with in vitro findings. Interestingly, LAT1 expression was retained following both doxorubicin and ^211^At-AAMT treatment (Appendix A), suggesting the potential of repeated ^211^At-AAMT therapy for tumor recurrence.

## 3. Discussion

In this study, we employed both in vitro and in vivo models to investigate the therapeutic efficacy and safety of ^211^At-AAMT, a LAT1-targeted alpha-releasing compound, in bone and soft tissue sarcomas. Although these malignancies are rare, they present significant therapeutic challenges owing to their resistance to conventional chemotherapy and radiation therapy as well as limited treatment options for unresectable and metastatic cases [17]. The findings presented in this study suggest that ^211^At-AAMT may serve as an effective and minimally invasive treatment for this disease.

^211^At-labeled agents have been investigated in a variety of malignancies, including glioblastoma, pancreatic cancer, prostate cancer, differentiated thyroid cancer, neuroblastoma, and pheochromocytoma, and have demonstrated significant antitumor effects, suggesting their potential clinical applications [18,19,20,21,22,23]. In Japan, multiple clinical trials are currently underway for various cancer types involving ^211^At-labeled compounds targeting disease-specific molecular transporters. For example, [^211^At] NaAt, which targets the sodium/iodide symporter, is been under investigation in a human phase I clinical trial for advanced differentiated thyroid cancer at Osaka University Hospital since November 2021 [24]. In addition, [^211^At] meta-astatobenzylguanidine, which is taken up via the norepinephrine transporter, is being evaluated at Fukushima Medical University in patients with malignant pheochromocytomas and paraganglioma [25]. In June 2024, a clinical trial of the prostate-specific membrane antigen (PSMA)-targeted radiotherapeutic agent [^211^At] PSMA-5 was initiated in patients with castration-resistant prostate cancer [26]. These studies highlight the growing promise of alpha-particle therapy in Japan and reinforce the potential of ^211^At-based targeted approaches for treatment-resistant cancers.

Several reports have described the application of alpha-particle radiation therapy to synovial sarcoma (SS) using ^211^At and ^225^Ac [27,28]. Li et al. reported that ^211^At-anti-Frizzled homolog 10 (FZD10) antibodies efficiently suppressed the growth of SS xenografts [27]. FZD10—a transmembrane protein member of the Frizzled family that serves as a putative receptor in the Wnt signaling pathway—is highly expressed in SS tumors but largely absent in other types of sarcomas and most normal tissues, suggesting its potential as a promising subtype-specific molecular target [27]. However, to date, no studies have reported the use of alpha therapy in other types of sarcomas, highlighting the need for further investigation in this area. In the present study, we demonstrated that LAT1 is overexpressed in multiple sarcoma subtypes and confirmed the antitumor efficacy of LAT1-targeted alpha therapy, which may be broadly applicable across diverse sarcoma subtypes. Furthermore, the small-molecule LAT1-targeted agent ^211^At-AAMT offers several advantages over antibody-targeted alpha therapies, including high specificity and affinity, favorable tumor permeability with rapid accumulation, and rapid clearance from most normal tissues [29,30]. Additionally, the relatively simple chemical structure of AAMT facilitates efficient radiolabeling and scalable production, both of which are important for clinical applications [16].

Fujimoto et al. also reported high LAT1 expression in tumor tissues from patients with clear cell sarcoma (CCS) and demonstrated that boron neutron capture therapy using LAT1-mediated uptake of p-borono-l-phenylalanine achieved complete local control of CCS [31]. This clinical case report further supports the potential utility of LAT1-target radiotherapy for bone and soft tissue sarcomas.

In this study, immunohistochemical (IHC) and immunofluorescence analyses showed that ^211^At-AAMT induced significant DSBs (increased γH2AX foci) both in vitro and in vivo. The potent DNA-damaging effect of alpha particles, coupled with their selective delivery to LAT1-expressing tumor cells, may contribute to the tumor suppressive effect. Notably, the antiproliferative effect persisted after a single administration, suggesting the possibility of biological effects beyond the physical half-life of ^211^At (half-life 7.2 h), such as the induction of tumor cell senescence or remodeling of the tumor microenvironment [32]. Importantly, LAT1 expression persisted in residual tumor tissues following ^211^At-AAMT treatment, suggesting the potential for re-irradiation or combination therapy. Further investigation is warranted to assess the radiosensitivity of LAT1-expressing cells after ^211^At-AAMT treatment.

Although the adverse events associated with ^211^At-based alpha therapy in humans have not yet been comprehensively reported, previous toxicity studies of [^211^At]PSMA-5 demonstrated transient changes, with no irreversible toxicity observed 14 days after administration [18]. Consistently, our previous toxicity study of ^211^At-AAMT, with follow-up for 28 days, showed transient hematopoietic suppression that subsequently recovered, with no pathological abnormalities in major organs [16]. Notably, in the present study, although doxorubicin induced significant weight loss at effective doses, a single dose of ^211^At-AAMT suppressed tumor growth without causing weight loss.

Despite these promising results, this study has some limitations. First, the xenograft model used in this study did not fully recapitulate the complex tumor microenvironment and immune system of human sarcomas, and thus the findings, including the retention of LAT1 expression after ^211^At-AAMT treatment, may not be entirely extrapolated to clinical settings. Confirmation in more advanced preclinical systems, such as patient-derived xenografts, will be necessary. Second, the therapeutic sensitivity of ^211^At-AAMT varied among sarcoma cell lines, and this variability could not be explained solely by LAT1 expression levels. Other biological factors—including the functional activity of the LAT1–CD98 complex, intracellular retention and efflux of radiolabeled compounds, proliferative capacity, and DNA damage repair proficiency—may contribute. Further experimental studies are required to elucidate these mechanisms. Third, to more firmly establish the LAT1 specificity of ^211^At-AAMT, additional studies using shLAT1 sarcoma cells are warranted, as these models will enable a direct demonstration of the causal link between LAT1 expression and therapeutic efficacy. Fourth, although a significant effect was demonstrated with a single dose, the optimal dosing schedule, including the possibility of divided dosing, remains to be established. Fifth, comprehensive toxicity assessments, including off-target effects at LAT1 expression sites in normal tissues such as the blood–brain barrier and placenta, as well as long-term follow-up to detect potential delayed adverse events, are essential prior to clinical application.

## 4. Materials and Methods

### 4.1. Production and Isolation of ^211^At

^211^At was produced via α-particle irradiation of bismuth-209 targets using an AVF cyclotron at the Research Center for Nuclear Physics, Osaka University (Ibaraki, Japan), through a nuclear reaction ^209^Bi(α,2n) ^211^At [33]. The bismuth layer, deposited onto an aluminum substrate, was irradiated with an α-beam, and ^211^At was subsequently isolated from the irradiated target via dry distillation.

Dry distillation and purification were performed using COSMiC-Mini VTRSC2 (Nihon Mediphysics Business Support, Hyogo, Japan), an automated distillation system, and the purified ^211^At was obtained as an aqueous solution. This system was developed based on previously reported dry separation protocols [34].

### 4.2. Synthesis of 211At-AAMT-O-Me-L

^211^At-AAMT-O-Me-L was synthesized via the Shirakami reaction using a boron-containing precursor (AAMT-O-Me-L) following the method described by Kaneda et al. [16]. The precursor was custom-synthesized by Kishida Chemical Co., Ltd. (Osaka, Japan). For precursors containing a pinacolborane moiety, a 7% sodium bicarbonate solution (Meylon^®^, Otsuka Pharmaceutical, Tokyo, Japan) was used as the reaction solvent, whereas water was used as a boronic acid derivative. For radiolabeling, approximately 10 MBq of aqueous ^211^At was mixed with the precursor compound and potassium iodide as the carrier. The reaction was performed at 50 °C for 50 min. The crude product was purified using an Oasis HLB column (Waters, Milford, MA, USA). The radiochemical purity and identity of the labeled compounds were confirmed by HPLC and TLC. TLC analysis was performed on silica gel G60 plates (Merck Millipore, Burlington, MA, USA) using a mobile phase of n-butanol/acetic acid/water (4:1:1). Radiolabeled spots were visualized using the Typhoon FLA-7000 biomolecular imager (GE Healthcare, Milwaukee, WI, USA). The final ^211^At-AAMT-O-Me-L product was adjusted to a concentration of 5 MBq/mL and mixed with 1.0% (*w*/*v*) ascorbic acid (pH 6.0), which served as reducing and stabilizing agents, respectively.

### 4.3. Cell Culture

The human cell line Kitra-SRS (CIC-DUX4sarcoma; RRID:CVCL_YI69) was established in our laboratory. MP-CCS-SY (CCS; RRID:CVCL_0J33) and SYO-1 (synovial sarcoma; RRID:CVCL_7146) were kindly provided by Dr. Moritake (Miyazaki University, Miyazaki, Japan) and Dr. Ozaki (Okayama University, Okayama, Japan), respectively. VAESBJ (epithelioid sarcoma; RRID:CVCL_1785), HT-1080 (fibrosarcoma; RRID:CVCL_0317), 143B (osteosarcoma; RRID:CVCL_2270), MG-63 (osteosarcoma; RRID:CVCL_0426), and U2OS (osteosarcoma; RRID:CVCL_0042) were purchased from ATCC. Saos-2 cells (osteosarcoma; RRID:CVCL_0548) were obtained from the Riken Cell Bank. NHDFs were purchased from Kurabo (Osaka, Japan). All cell lines were maintained in DMEM (Nacalai Tesque, Tokyo, Japan), which was supplemented with 10% FBS (Sigma-Aldrich, St. Louis, MO, USA) and 1% penicillin–streptomycin (100 IU/mL of penicillin and 100 μg/mL of streptomycin) at 37 °C in a humidified incubator with 5% CO2. Cell line authentication was performed based on morphological assessments, PCR-based genotyping, and the evaluation of growth characteristics. All experiments used cells between passages 10 and 30. Prior to experimentation, mycoplasma contamination was routinely tested and confirmed to be negative using the PCR Mycoplasma Detection Set (Takara Bio Inc., Shiga, Japan).

### 4.4. Western Blotting

For lysate preparation, cultured cells were washed with PBS and lysed using RIPA lysis and an extraction buffer (Thermo Fisher Scientific, Waltham, MA, USA) supplemented with a 1% protease and phosphatase inhibitor cocktail (Cell Signaling Technology). Tumor tissues were homogenized and lysed using the T-PER Tissue Protein Extraction Reagent (Thermo Fisher Scientific). Protein concentrations were determined using a bicinchoninic acid (BCA) assay (Thermo Fisher Scientific).

Equal amounts of protein lysates were separated by SDS-PAGE using 4–12% Bis-Tris gels (Life Technologies, Waltham, MA, USA) and transferred onto PVDF membranes (Nippon Genetics, Tokyo, Japan). The membranes were blocked with 5% skim milk in TBS with Tween 20 at room temperature and incubated overnight at 4 °C, with the primary antibodies diluted in Can Get Signal Solution 1 (TOYOBO, Osaka, Japan). After washing, the membranes were incubated for 1 h at room temperature, with the secondary antibodies diluted in Can Get Signal Solution 2 (TOYOBO). Protein bands were visualized using the ChemiDOC Touch Imaging System (Bio-Rad, Hercules, CA, USA). The primary antibodies used in the experiments are listed in Appendix A.

Western blot images were acquired as 8-bit TIFFs, and band intensities of LAT1 and β-actin were quantified using Fiji/ImageJ (version 1.54p, National Institutes of Health, Bethesda, MD, USA). LAT1 signals were background-corrected and normalized to the corresponding β-actin band. For soft tissue sarcoma cell lines, values were expressed relative to HT-1080 (=1.0) within each blot, and for osteosarcoma cell lines, values were expressed relative to U-2 OS (=1.0).

### 4.5. Cellular Uptake Assay

To investigate the cellular uptake of ^211^At-AAMT via LAT1, various cell lines were seeded in 24-well plates at a density of 2.5 × 10^3^ cells per well and cultured for 48 h. Following 30 min incubation in HBSS, the cells were treated with 1 kBq of ^211^At-AAMT in the presence or absence of 200 mmol/L of the dual LAT1/LAT2 inhibitor BCH (Sigma-Aldrich) and 10 µmol/L of the LAT1-selective inhibitor nanvuranlat (JPH203; GlpBio Technology Inc., Montclair, CA, USA). The cells were subsequently washed twice with PBS(−) and lysed in 0.1 N NaOH, with the radioactivity measured using a gamma counter (2480 Wizard^2^; PerkinElmer, Waltham, MA, USA). The total protein content was quantified using a BCA protein assay kit (FUJIFILM Wako Pure Chemical Corporation, Osaka, Japan) and a microplate reader.

### 4.6. Cell Proliferation Assay

Cells were seeded in 96-well plates at a density of 2 × 10^3^ cells per well in triplicates. After overnight incubation, the cells were treated with either ^211^At-AAMT or doxorubicin hydrochloride (TCI, Tokyo, Japan) for the indicated durations. The vehicle control for ^211^At-AAMT was its formulation buffer (0.2% acetic acid with 1.0% ascorbic acid, pH 6.0), whereas DMSO was used as the vehicle control for doxorubicin.

Cell proliferation was evaluated using the WST-8 assay with Cell Count Reagent SF (Nacalai Tesque) following the manufacturer’s instructions. The absorbance was measured at 450 and 690 nm (reference wavelength) using a spectrophotometer (MultiskanTM FC, Thermo Fisher Scientific). Relative proliferation rates were calculated by normalizing the absorbance values to the average of each respective control group, which was set to 1. IC50 values were log-transformed for statistical analysis; geometric means and 95% confidence intervals are reported.

### 4.7. Colony Formation Assay

Cells were seeded in 24-well plates at a density of 1 × 10^3^ cells per well in quadruplicates. After overnight incubation, the cells were treated with various doses of ^211^At-AAMT or doxorubicin for 48 h, followed by replacement with a fresh medium. When the control cells reached approximately 80% confluence, all wells were washed twice with PBS(−) and stained with 0.2% crystal violet for 30 min. The plates were then washed with water and photographed. For quantification, the dye was extracted using a solution of 50% ethanol and 0.02 mol/L hydrochloric acid, and the absorbance was measured at 540 nm using a spectrophotometer.

### 4.8. Cellular Immunofluorescence Staining

Cells were seeded in an 8-well slide chamber (WATSON Slide Chamber 8 Well; WATSON Co., Ltd., Tokyo, Japan) and incubated overnight to allow cell attachment. The next day, the cells were treated with ^211^At-AAMT for 24 h. After treatment, the cells were washed twice with PBS and fixed with 4% paraformaldehyde at room temperature for 30 min. The cells were permeabilized with 0.1% Triton X-100 in PBS for 15 min and blocked with 3% BSA in PBS for 30 min.

The cells were incubated overnight at 4 °C with a rabbit monoclonal anti-phospho-histone γH2AX (Ser139) antibody (clone 20E3, Cell Signaling Technology, #9718) diluted at 1:800 in a blocking buffer. The following day, the cells were washed three times with PBS and incubated for 2 h at room temperature in the dark with the Alexa Fluor 594-conjugated goat anti-rabbit IgG (H + L), F(ab’)_2_ fragment (Cell Signaling Technology, #8889, 1:1000 dilution). After staining, the upper part of the slide chamber was removed, and the slides were mounted using ProLong^TM^ Gold Antifade Mountant with DAPI (Thermo Fisher Scientific, #S36939). Fluorescence images were acquired using a BZ-X810 fluorescence microscope (Keyence Corporation, Osaka, Japan).

For quantification, γH2AX fluorescence intensity per nucleus was measured using Fiji/ImageJ. Regions of interest (ROIs) were defined around individual nuclei based on DAPI staining, and the integrated density was calculated. Background correction was performed by subtracting the product of the ROI area and the mean background intensity. The corrected integrated density values were normalized to the 0 kBq control (=1). For foci analysis, γH2AX foci were manually counted 50 nuclei per condition, and the average number of foci per nucleus was calculated.

### 4.9. In Vivo Xenograft Models

Five-week-old female BALB/c nu/nu athymic mice (n = 64, body weight range: 17.1–22.6 g) were housed under standard laboratory conditions at the Institute of Experimental Animal Sciences of Osaka University Medical School. All animal experiments were performed in accordance with the protocols approved by the Institutional Animal Care and Use Committee of the Graduate School of Medicine, Osaka University Graduate School of Medicine. To establish subcutaneous xenograft tumor models, 1 × 10^7^ MP-CCS-SY or 143B cells were injected into the left dorsal flank of each mouse. When the tumors reached an average volume of approximately 200 mm^3^, treatment was initiated. In the MP-CCS-SY model, mice were randomly assigned to four groups (n = 5 per group): 3 mg/kg or 6 mg/kg doxorubicin, 1 MBq of ^211^At-AAMT, or vehicle control. By contrast, the 143B model included three groups (n = 5 per group): 3 mg/kg doxorubicin, 1 MBq of ^211^At-AAMT, and vehicle control. The 6 mg/kg doxorubicin group was not used in this model. Doxorubicin was administered intraperitoneally every 4 days, whereas ^211^At-AAMT was administered intravenously via the tail vein. The tumor size and body weight were measured every alternate day. Tumor volumes were calculated using caliper measurements according to the following formula: tumor volume (mm^3^) = (length × width^2^)/2. When the tumor volume exceeded 2000 mm^3^, the mice were humanely euthanized by CO_2_ asphyxiation and tumors were excised and weighed. To assess histological changes 24 h post-treatment, MP-CCS-SY and 143B xenograft models (n = 4 per group) were treated with 3 mg/kg doxorubicin or 1 MBq ^211^At-AAMT, followed by tissue collection for histological analysis.

### 4.10. Immunohistochemistry

Excised xenograft tumors were fixed in 10% neutral-buffered formalin and processed for paraffin embedding. Serial sections were prepared at a thickness of 3.5 μm and subjected to histological evaluation by hematoxylin and eosin (HE) staining and IHC analysis. For IHC staining, paraffin sections were deparaffinized and rehydrated using a graded alcohol series. Antigen retrieval was performed by heating the sections in 10 mM citrate buffer (pH 6.0) at 95 °C for 30 min. For LAT1-specific staining, antigen retrieval was performed by autoclaving the sections in HistoFine antigen retrieval solution (pH 9.0, Nichirei Biosciences, Tokyo, Japan) for 5 min. To block endogenous peroxidase activity, the sections were incubated with methanol containing 3% hydrogen peroxide for 10 min. Nonspecific binding was blocked by incubating the sections in TBS containing 2% BSA for 1 h at room temperature. Primary antibodies were then applied, and the slides were incubated overnight at 4 °C. The following day, the sections were incubated with HRP-conjugated secondary antibodies for 1 h. Color development was achieved using 3,3′-diaminobenzidine tetrahydrochloride (Dako, Carpinteria, CA, USA), followed by nuclear counterstaining with hematoxylin. The details of the primary antibodies used in this study are presented in Appendix A. All IHC analyses were performed using Aperio CS2 (Leica, Wetzlar, Germany), and staining intensities were compared at ×200. LAT1 expression was confirmed at ×50 for each preparation, and LAT1 expression was scored as follows: 0 = no expression; 1 = <10% positive cells; 2 = 10–30%, 3 = 30–50%, and 4 = >50% of tumor cells showing positive staining.

### 4.11. Statistical Analysis

All results are presented as means ± standard deviation (SD). Statistical comparisons for in vitro assays were performed using Student’s *t*-test, whereas differences in animal study data were evaluated using the Mann–Whitney U test. A two-tailed *p*-value of <0.05 was considered statistically significant.

## 5. Conclusions

LAT1 represents a promising therapeutic target for TAT across various sarcoma subtypes. ^211^At-AAMT shows strong potential as a therapeutic agent, demonstrating tumor suppression that is comparable to or more effective than conventional chemotherapy (e.g., doxorubicin) while exhibiting reduced toxicity. Further studies are needed to optimize dosing regimens, evaluate long-term safety, investigate combinations with conventional chemotherapy, and select patients based on LAT1 expression levels. LAT1-targeted TAT may offer an effective treatment modality for refractory sarcomas.

## Figures and Tables

**Figure 1 ijms-26-08599-f001:**
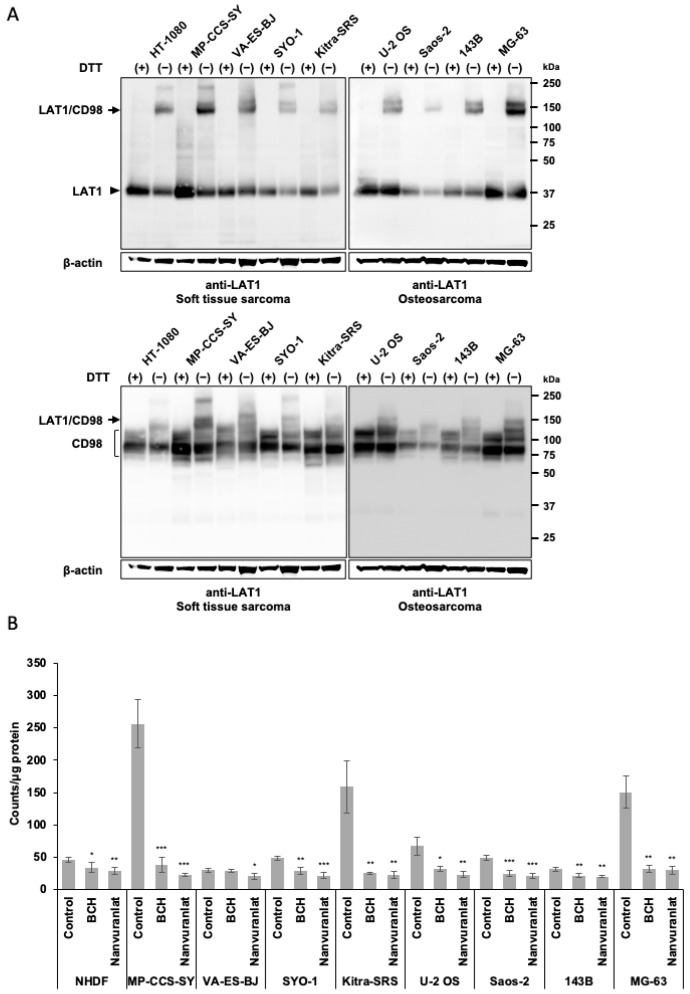
LAT1 expression and ^211^At-AAMT uptake in sarcoma cell lines. (**A**) Western blot analysis of LAT1 and CD98 in osteosarcoma and soft tissue sarcoma cell lines under reducing (+DTT) and non-reducing (−DTT) conditions. LAT1-CD98 heterodimer (150 kDa), LAT1 monomer (39 kDa), and CD98 (75–120 kDa) are indicated. β-actin served as a loading control. (**B**) Uptake of ^211^At-AAMT in sarcoma cell lines with or without BCH or nanvuranlat. Data are shown as means ± standard deviation (SD) (n = 4). * *p* < 0.05, ** *p* < 0.01, *** *p* < 0.001 compared with untreated controls. LAT1, L-type amino acid transporter 1; BCH, 2-aminobicyclo [2.2.1] heptane-2-carboxylic acid; ^211^At-AAMT, ^211^At-AAMT-O-Me-L.

**Figure 2 ijms-26-08599-f002:**
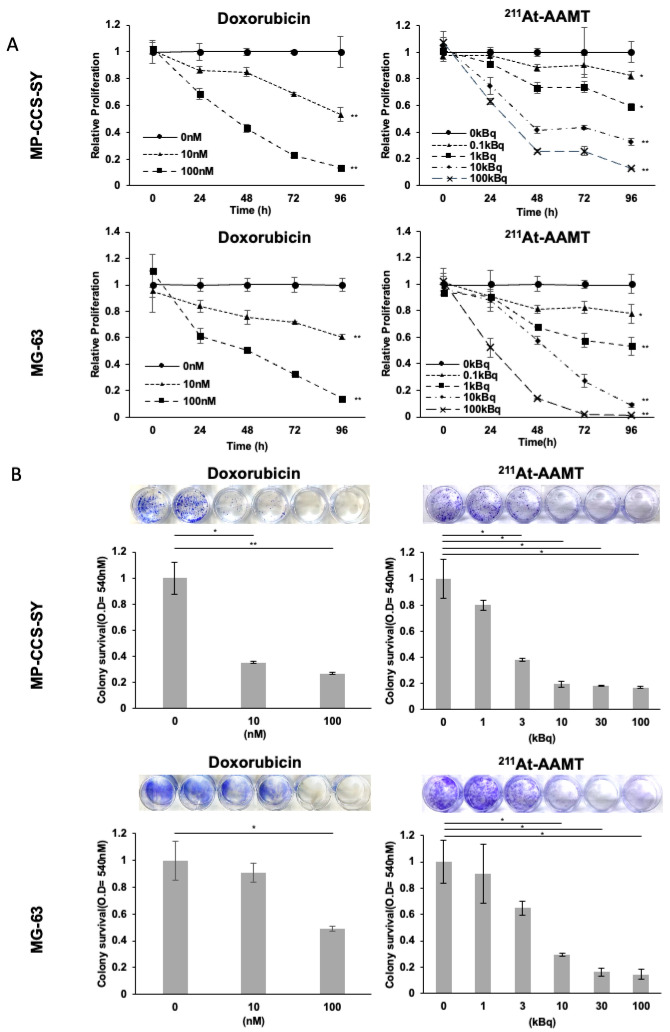
Growth inhibition and colony suppression by ^211^At-AAMT. (**A**) Dose-dependent inhibition of proliferation in MP-CCS-SY and MG-63 cells treated with ^211^At-AAMT (0, 0.1, 1, 10, and 100 kBq) or doxorubicin (0, 10, and 100 nM) for 48 h. Data are shown as means ± standard deviation (SD) (n = 3). * *p* < 0.05, ** *p* < 0.01 compared with the vehicle control. (**B**) Colony formation assay after 48 h treatment with ^211^At-AAMT or doxorubicin. Representative stained images and their quantification are shown. Data are shown as means ± standard deviation (SD) (n = 3). 3 kBq of ^211^At-AAMT significantly suppressed tumoral cell growth, which was comparable to that induced by 10 to 100 nM doxorubicin. * *p* < 0.05, ** *p* < 0.01 compared with the control. ^211^At-AAMT, ^211^At-AAMT-O-Me-L.

**Figure 3 ijms-26-08599-f003:**
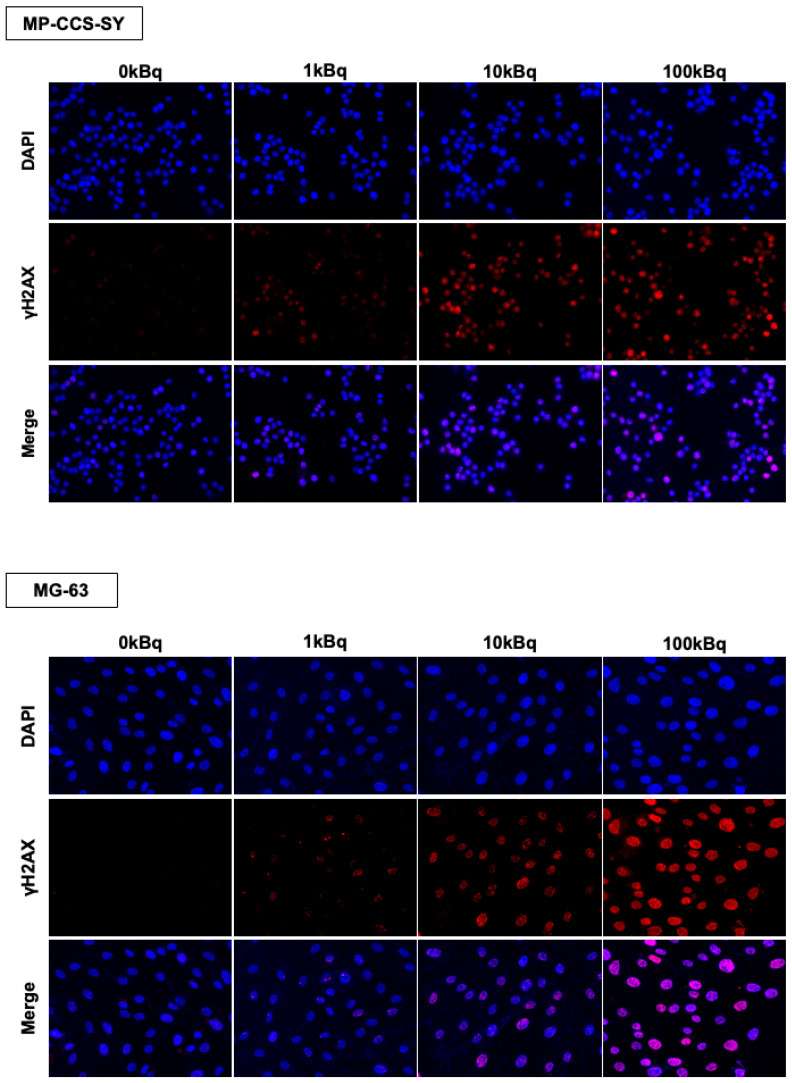
DNA double-strand break induction by ^211^At-AAMT. Immunofluorescence staining of γH2AX (red) and DAPI (blue) in MP-CCS-SY and MG-63 cells 24 h after treatment with ^211^At-AAMT. Merged images show γH2AX foci formation, indicative of DNA double-strand breaks. Images were acquired at 40× magnification. ^211^At-AAMT, ^211^At-AAMT-O-Me-L.

**Figure 4 ijms-26-08599-f004:**
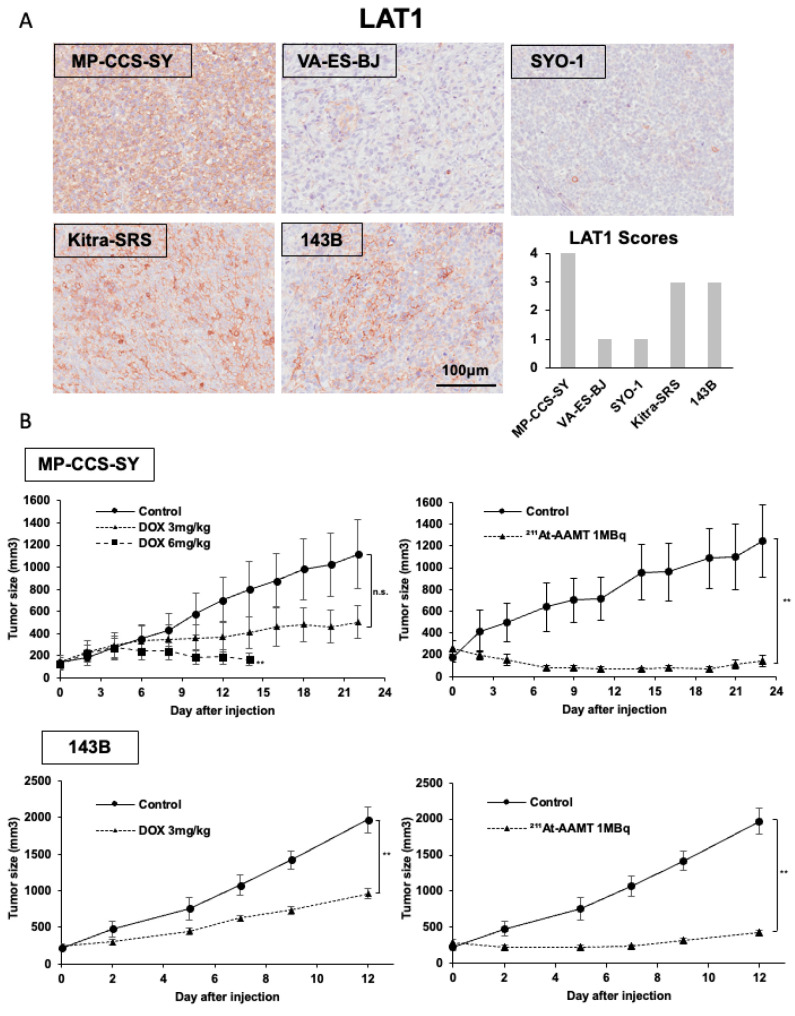
In vivo antitumor effects of ^211^At-AAMT in xenograft models. (**A**) LAT1 expression in xenograft tumor models (MP-CCS-SY, VA-ES-BJ, SYO-1, Kitra-SRS, and 143B cells), as assessed by immunohistochemistry. MP-CCS-SY tumors showed strong LAT1 expression (score: 4), whereas VA-ES-BJ and SYO-1 tumors showed low expression (score: 1), consistent with the in vitro results. (**B**) Tumor growth curves of xenograft-bearing mice treated with the vehicle, doxorubicin (3 or 6 mg/kg, intraperitoneally every 4 days), or ^211^At-AAMT (1 MBq, single intravenous injection). In the MP-CCS-SY model, both ^211^At-AAMT and 6 mg/kg doxorubicin significantly suppressed tumor growth. In the 143B model, ^211^At-AAMT also led to significant tumor suppression. Tumor volumes are shown as means ± standard error (SE) (n = 5). End-point tumor volumes were compared with the control using the Mann–Whitney U test; exact *p*-values are indicated in the Results. ** *p* < 0.01 compared with the control. LAT1, L-type amino acid transporter 1; ^211^At-AAMT, ^211^At-AAMT-O-Me-L.

**Figure 5 ijms-26-08599-f005:**
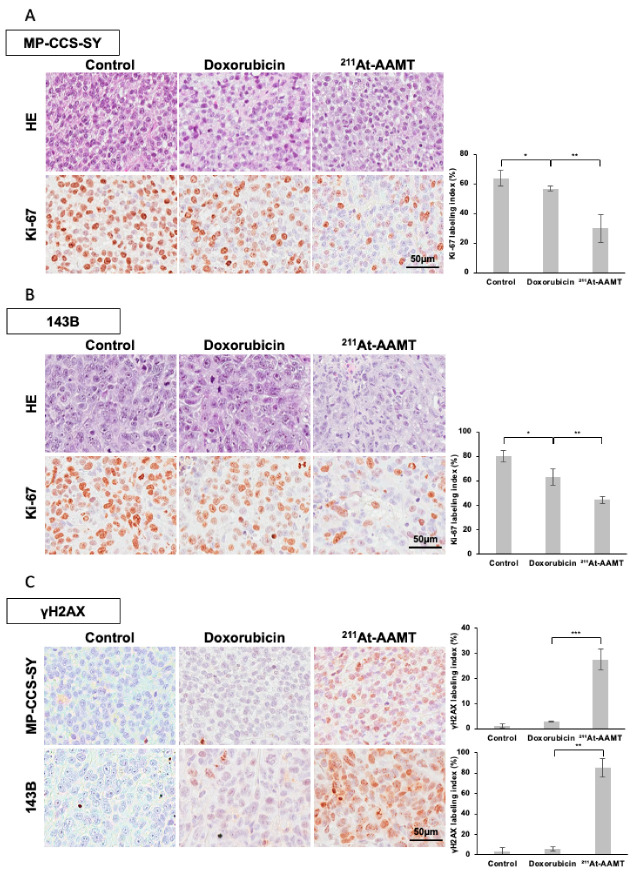
Histopathological and immunohistochemical analysis of treated tumors. HE staining and Ki-67 immunostaining of MP-CCS-SY (**A**) and 143B (**B**) xenograft tumor models collected at the experimental endpoint. The adjacent bar graph shows the percentage of Ki-67-positive cells for each treatment group. (**C**) γH2AX immunostaining of MP-CCS-SY and 143 B xenograft tumors 24 h after treatment with ^211^At-AAMT or doxorubicin. The bar graphs represent the percentage of γH2AX-positive cells in each group. Data are shown as means ± standard deviation (SD) ((**A**,**B**): n = 5; (**C**): n = 4). * *p* < 0.05, ** *p* < 0.01, and *** *p* < 0.001 compared with the control. ^211^At-AAMT, ^211^At-AAMT-O-Me-L.

## Data Availability

Data can be found within the article.

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
