# Peer review of "LAT1-Targeted Alpha Therapy Using 211At-AAMT for Bone and Soft Tissue Sarcomas"

_ijms, 2025, doi:10.3390/ijms26178599_

Round 1
Reviewer 1 Report
Comments and Suggestions for Authors
The authors explore the therapeutic potential of a LAT1 targeted alpha emitter radiopharmaceutical for sarcomas.
The research is well presented, the methods clearly described and the conclusions are promising.
However, this reviewer considers that adjustments need to be made to enhance the clarity and overall impact of the work. Detailed comments and questions that this reviewer considers the authors need to clarify/answer were highlighted and comments introduced in the attached pdf file.

Author Response
We sincerely appreciate the reviewer’s valuable comments and suggestions, which have helped us to improve our manuscript. Our responses are provided below.
Comments 1: This reviewer suggests changing soft tissue tumors to soft tissue sarcomas.
Not all soft tissue tumors are malignant, and the authors focus on sarcomas.
Response 1: We appreciate the reviewer’s helpful comment. We have revised the text by replacing “soft tissue tumors” with “soft tissue sarcomas” to accurately reflect the focus of our study, as follows:
LAT1-Targeted Alpha Therapy using 211At-AAMT for Bone and Soft Tissue Sarcomas
Page 1, Title
Comments 2: replace and by a comma in superscript
Response 2: We thank the reviewer for pointing this out. We have revised the superscript citations by replacing “and” with a comma as requested.
Kaneda-Nakashima2, 3
Page 1, line 5
Comments 3: Please indicate the tumoral type evaluated in this previous work.
A figure with the structure of AAMT would be informative
Response 3: We are grateful to the reviewer for this valuable suggestion. We have revised the text to indicate that a human pancreatic cancer cell line was used in the previous xenograft study. In addition, the chemical structure of 211At-AAMT-O-Me-L has been included in Supplementary Figure S1. The main text has been revised accordingly, as shown above.
The chemical structures of 211At-AAMT-OH-L and 211At-AAMT-O-Me-L are shown in Supplementary Figure S1. Preclinical xenograft studies using a human pancreatic cancer cell line have confirmed its efficacy in inhibiting tumor growth with limited accumulation in normal tissue [16].
Page 2, lines 78–81, Introduction, Supplementary Figure S1
Comments 4: Why did the authors did not use HT1080 as a positive control in the uptake studies?
Response 4: We apologize for not including HT-1080 as a positive control in the uptake study. In this experiment, we selected normal human dermal fibroblasts (NHDF) as a negative control to compare with tumor cells and therefore did not include HT-1080 as a positive control.
Comments 5: this seems to be the case for MP-CCS-SY but not for MG-63. Please clarify
Response 5: We thank the reviewer for this comment. As correctly pointed out, this finding applies only to MP-CCS-SY cells. We have therefore revised the text as follows:
In MP-CCS-SY cells, the inhibitory effect of 3 kBq 211At-AAMT appeared comparable to that of 10 nM doxorubicin (Figure 2B).
Page 4, lines 128–129, Results
Comments 6: replace tumor growth with tumoral cell growth
Response 6: We appreciate the reviewer’s suggestion. We have revised the text by replacing “tumor growth” with “tumoral cell growth” as suggested.
3 kBq of 211At-AAMT significantly suppressed tumoral cell growth, which was comparable to that induced by 10 to 100 nM doxorubicin.
Page 4, line 137, Figure 2
Comments 7: Did the authors quantify the DNA damage (either number of foci or total fluorescent intensity)?
Response 7: We thank the reviewer for this valuable comment. In response, we quantified DNA damage using two complementary approaches: (i) γH2AX fluorescence intensity per nucleus was measured, background-corrected, and normalized to the 0 kBq control (=1), and (ii) the number of γH2AX foci per nucleus was quantified. The quantitative results, presented as dose–response bar graphs with mean ± SD, are now included in Supplementary Figure S3. A statistically significant increase in fluorescence intensity was observed in MP-CCS-SY cells at 100 kBq compared with 0 kBq (p = 0.03), whereas MG-63 cells showed a similar trend without reaching statistical significance. In addition, foci quantification demonstrated a dose-dependent increase in the mean number of γH2AX foci per cell. We have also added a description of these quantification procedures to the Methods section. The Results and Methods sections have been revised as follows:
Moreover, 211At-AAMT treatment induced DNA damage owing to the formation of DNA double-strand breaks (DSBs) (Figure 3), as evidenced by a significant increase in the total fluorescence intensity of γH2AX in the 100 kBq group relative to the 0 kBq group (Figure S3A), and by a dose-dependent increase in the mean number of γH2AX foci per cell (Figure S3B).
Page 5, lines 139–143, Results
For quantification, γH2AX fluorescence intensity per nucleus was measured using Fiji/ImageJ. Regions of interest (ROIs) were defined around individual nuclei based on DAPI staining, and the integrated density was calculated. Background correction was performed by subtracting the product of the ROI area and the mean background intensity. The corrected integrated density values were normalized to the 0 kBq control (=1). For foci analysis, γH2AX foci were manually counted 50 nuclei per condition, and the average number of foci per nucleus was calculated.
Page 12, lines 402–408, Materials and Methods
Supplementary Figure S3
Comments 8: An introductory sentence, explaining the different models produced is missing.
Response 8: We thank the reviewer for this helpful comment and apologize for the omission. We have added an introductory sentence explaining the different xenograft models produced, as follows:
To investigate the in vivo antitumor effects of 211At-AAMT, we established xenograft models using human sarcoma and osteosarcoma cell lines with different levels of LAT1 expression.
Page 6, lines 153–155, Results
Comments 9: Why not the Kitra-SRS that showed a higher expression in vitro (fig1) and also in the ex vivo analysis?
Response 9: We appreciate the reviewer’s comment. In our densitometric analysis of Western blot bands normalized to β-actin (Supplementary Table S1), MP-CCS-SY exhibited the highest LAT1 expression among the malignant soft tissue sarcoma cell lines both in vitro and in ex vivo xenograft tumors. Therefore, MP-CCS-SY was selected as the representative soft tissue sarcoma model for in vivo studies. For osteosarcoma, we chose the 143B cell line, which demonstrated high LAT1 expression and readily formed tumors in vivo. Although Kitra-SRS also showed relatively high LAT1 expression, it did not surpass MP-CCS-SY in the quantitative analysis, and thus we prioritized MP-CCS-SY for subsequent in vivo experiments.
Comments 10: A similar experiment but with incubation the LAT1 inhibitors would be useful to assess specificity in vivo
Response 10: We greatly appreciate the reviewer’s insightful comment. The specificity of LAT1-targeted alpha therapy with ²¹¹At-AAMT is indeed an important issue. To more rigorously evaluate this specificity, we are currently generating shLAT1 sarcoma cells. These experiments are ongoing, and the results will be reported in a subsequent study. In addition, this point has now been reflected in the revised manuscript as part of the Limitation section, as follows:
Third, to more firmly establish the LAT1 specificity of 211At-AAMT, additional studies using shLAT1 sarcoma cells are warranted, as these models will enable a direct demonstration of the causal link between LAT1 expression and therapeutic efficacy.
Page 9, lines 278–281, Discussion

Reviewer 2 Report
Comments and Suggestions for Authors
The submitted manuscript describes a very thorough in vitro study on the therapeutic efficacy of an alpha emitter in the chemical form of a small molecule.
Line 60: “high energy emission”, please specify as “high energy alpha emission”
Line 66-78: A drawn representation of the chemical structure instead of the long abbreviation, which is then shortened for simplicity, could completely avoid misunderstandings about the actual compound being studied. Moreover, a previous publication uses the same abbreviation for a different compound. In reference 15 211At-AAMT is used for 211At-AAMT-OH-L, where in ref 16 the same short abbreviation 211At-AAMT means the O-methylated compound 211At-AAMT-OMe-L. This is quite confusing.
Line 108-109: 211At-AAMT as 3-[211At] Astato-alpha-methyl-L-tyrosine, again confusing: O-Methylated or not? 3-Astato-alpha-methyl-L-O-methyltyrosine?
Line 261: “through a nuclear reaction 209Bi(α,2n) between 211At”
following points:
• What is the main question addressed by the research?
The main question is: Is targeted alpha therapy (TAT) with small molecules a promising alternative to chemotherapy?
• Do you consider the topic original or relevant to the field?Yes, i consider the topic original and relevant for the field.
Does itaddress a specific gap in the field? Please also explain why this is/ is not
the case.
Every new potentially therapeutic compound must be tested in this or a similar way before extremely complex and expensive clinical trials can begin.
• What does it add to the subject area compared with other publishedmaterial?
It contributes significantly to the expansion of future treatment options
• What specific improvements should the authors consider regarding themethodology?
As already mentioned, the description of the chemical structure of 211At-AAMT could be improved
• Are the conclusions consistent with the evidence and arguments presentedand do they address the main question posed? Please also explain why this
is/is not the case.
The advantage of the TAT with LAT1 as therapeutic target and 211At-AAMT as therapeutic agent with significantly more effectiveness and fewer side effects compared to conventional chemotherapy could be demonstrated, the means the conclusion is consistent with the evidence.
• Are the references appropriate?Yes
• Any additional comments on the tables and figures.No need, the tables and figures are completely fine!
Author Response
We sincerely appreciate the reviewer’s valuable comments and suggestions, which have helped us to improve our manuscript. Our responses are provided below.
Comments 1: Line 60: “high energy emission”, please specify as “high energy alpha emission” 
Response 1: We thank the reviewer for this helpful comment. We have revised the text by replacing “high energy emission” with “high energy alpha emission” as suggested.
Among the radionuclides being studied for TAT, astatin-211 (211At) has received considerable attention owing to its adequate half-life of 7.2 h, high energy alpha emission, and feasibility of production using accelerator-based technology [11].
Page 2, lines 60–61, Introduction
Comments 2: Line 66-78: A drawn representation of the chemical structure instead of the long abbreviation, which is then shortened for simplicity, could completely avoid misunderstandings about the actual compound being studied. Moreover, a previous publication uses the same abbreviation for a different compound. In reference 15 211At-AAMT is used for 211At-AAMT-OH-L, where in ref 16 the same short abbreviation 211At-AAMT means the O-methylated compound 211At-AAMT-OMe-L. This is quite confusing.
Response 2: We thank the reviewer for pointing out the potential confusion caused by the use of abbreviations, and we apologize for the lack of clarity. To address this issue, we have first added Supplementary Figure S1, where the chemical structures of 211At-AAMT-OH-L and 211At-AAMT-O-Me-L are shown, in order to avoid any misunderstanding regarding the compounds studied.
The chemical structures of 211At-AAMT-OH-L and 211At-AAMT-O-Me-L are shown in Supplementary Figure S1.
Page 2, lines 78–79, Introduction, Supplementary Figure S1
In addition, we have revised the text to specify that in this study we used 211At-AAMT-O-Me-L, as follows:
In this study, we aimed to evaluate the antitumor effects of this novel TAT (211At-AAMT-O-Me-L) in malignant bone and soft tissue tumor cells compared with first-line chemotherapeutic agents currently used for bone and soft tissue sarcomas.
Page 2, lines 82–84, Introduction
Comments 3: Line 108-109: 211At-AAMT as 3-[211At] Astato-alpha-methyl-L-tyrosine, again confusing: O-Methylated or not? 3-Astato-alpha-methyl-L-O-methyltyrosine?
Response 3: We apologize for the multiple typographical errors in the description of the compound. We have corrected these instances to “211At-AAMT, 211At-AAMT-O-Me-L”
Page 3, lines 113–114, Figure 1
Page 4, line 138, Figure 2
Page 5, lines 150–151, Figure 3
Page 6, lines 180–181, Figure 4
Page 7, lines 202–203, Figure 5
Supplementary Figure S2
Supplementary Figure S3
Supplementary Figure S4
Supplementary Figure S5
Comments 4: Line 261: “through a nuclear reaction 209Bi(α,2n) between 211At”
Response 4: We thank the reviewer for this helpful comment. As pointed out, we have deleted “between” to correct the sentence.
211At was produced via α-particle irradiation of bismuth-209 targets using an AVF cyclotron at the Research Center for Nuclear Physics, Osaka University (Ibaraki, Japan), through a nuclear reaction 209Bi(α,2n) 211At [33].
Page 9, line 291, Materials and Methods
Reviewer 3 Report
Comments and Suggestions for Authors
The manuscript presents robust data on the efficacy of LAT1-targeted alpha therapy using 211At-AAMT in both bone and soft tissue sarcoma models, with well-structured in vitro and in vivo experiments. However, several minor points could be addressed to further enhance the clarity and rigor of the results. In the section describing LAT1 expression in cell lines, it would be helpful to quantitatively specify the relative expression levels observed in each cell type, indicating whether values are derived from densitometric analysis of Western blot bands or another method, and to provide means and standard deviations where appropriate. When discussing the inhibition of proliferation and colony formation, the systematic reporting of IC50 values and their confidence intervals for both doxorubicin and 211At-AAMT is recommended; additionally, the discussion should briefly consider whether differences in sensitivity among cell lines could be attributed to variations in LAT1 expression or other biological features. Regarding the assessment of DNA damage (γH2AX), please clarify whether foci analysis was performed quantitatively and include the mean number of foci per cell, standard deviations, and the number of cells analyzed, rather than relying solely on representative images. For the in vivo experiments, it is important to clearly state the number of animals used per group in each assay, and when presenting tumor growth curves, to specify the exact statistical tests applied (e.g., repeated measures ANOVA, Mann–Whitney test) and to report exact p-values for key comparisons, as sometimes only asterisks are shown. It would also be valuable to briefly comment on whether any additional adverse events were observed apart from body weight changes, such as organ toxicity, and to indicate whether the follow-up period was sufficient to detect potential delayed effects. In the histopathology section, while the inclusion of representative images is appreciated, a quantitative analysis of the proportion of Ki-67- and γH2AX-positive cells, together with standard deviations and sample sizes, would reinforce the robustness of the findings. Throughout the results section, please standardize the reporting of numerical values and units, clarify statistical significance thresholds, and always indicate the sample size (n) in figure legends and tables. Finally, in interpreting the results, it would be helpful to explicitly acknowledge the limitations of the xenograft models, noting that results from immunodeficient mice may not fully translate to the human setting, and that the retention of LAT1 expression after 211At-AAMT treatment should be confirmed in more complex models or additional preclinical studies. These detailed clarifications and expansions will further strengthen the clarity, reproducibility, and scientific rigor of the manuscript prior to publication.
Author Response
We sincerely appreciate the reviewer’s valuable comments and suggestions, which have helped us to improve our manuscript. Our responses are provided below.
Comment 1: In the section describing LAT1 expression in cell lines, it would be helpful to quantitatively specify the relative expression levels observed in each cell type, indicating whether values are derived from densitometric analysis of Western blot bands or another method, and to provide means and standard deviations where appropriate.
Response 1: We thank the reviewer for this helpful comment. In response, we have performed densitometric analysis of LAT1 protein expression using Fiji/ImageJ. Band intensities were background-corrected, normalized to the corresponding β-actin band, and expressed relative to HT-1080 (=1.0) for soft tissue sarcoma cell lines and U-2 OS (=1.0) for osteosarcoma cell lines within each blot. The quantitative results are now presented as mean ± SD in Supplementary Table S1 and are described in the revised Results section, as follows:
Densitometric quantification of LAT1–CD98 heterodimer bands, normalized to β-actin, revealed variable expression levels across cell lines (Table S1)
Page 3, lines 93–95, Results, Supplementary Table S1
In addition, we have added a description of the densitometric analysis procedure in the Materials and Methods section, as follows:
Western blot images were acquired as 8-bit TIFFs, and band intensities of LAT1 and β-actin were quantified using Fiji/ImageJ (version 1.54p, National Institutes of Health, Bethesda, MD, USA). LAT1 signals were background-corrected and normalized to the corresponding β-actin band. For soft tissue sarcoma cell lines, values were expressed relative to HT-1080 (=1.0) within each blot, and for osteosarcoma cell lines, values were expressed relative to U-2 OS (=1.0).
Pages 10–11, lines 348–353, Materials and Methods
Comment 2: When discussing the inhibition of proliferation and colony formation, the systematic reporting of IC50 values and their confidence intervals for both doxorubicin and 211At-AAMT is recommended.
Response 2: We thank the reviewer for this valuable comment. In accordance with the recommendation, we have reanalyzed the data and now report IC50 values together with their 95% confidence intervals. The revised values have been included in the Results section and Supplementary Figure S2, as follows:
The IC50 values at 48 h were comparable between the two cell lines. For doxorubicin, the IC50 was 31.0 nM (95% CI, 21.3–45.0; n=3) in MP-CCS-SY cells and 70.7 nM (95% CI, 44.2–113.0; n=3) in MG-63 cells. For ²¹¹At-AAMT, the IC50 was 100.8 kBq (95% CI, 37.5–271.0; n=3) in MP-CCS-SY cells and 88.8 kBq (95% CI, 16.5–478.8; n=3) in MG-63 cells (Figure S2).
Page 4, lines 122–126, Results
Supplementary Figure S2
In addition, the Materials and Methods section was revised as follows:
IC50 values were log-transformed for statistical analysis; geometric means and 95% confidence intervals are reported.
Page 11, lines 375–376, Materials and Methods
Comment 3: additionally, the discussion should briefly consider whether differences in sensitivity among cell lines could be attributed to variations in LAT1 expression or other biological features.
Response 3: We sincerely thank the reviewer for this insightful comment. In our study, although LAT1 expression and LAT1-mediated uptake of 211At-AAMT were confirmed in all tested sarcoma cell lines, the differences in sensitivity did not correlate directly with LAT1 expression levels in vitro. We therefore consider that, in addition to LAT1 abundance, other biological factors—including functional activity of the LAT1–CD98 complex, intracellular retention and efflux of radiolabeled compounds, proliferative capacity, and DNA damage repair proficiency—may underlie the observed variability in therapeutic response. Further experimental studies will be required to elucidate these mechanisms in future work. In addition, this point has now been incorporated into the revised manuscript as part of the Limitation section, as follows:
Second, the therapeutic sensitivity of 211At-AAMT varied among sarcoma cell lines, and this variability could not be explained solely by LAT1 expression levels. Other biological factors—including the functional activity of the LAT1–CD98 complex, intracellular retention and efflux of radiolabeled compounds, proliferative capacity, and DNA damage repair proficiency—may contribute. Further experimental studies are required to elucidate these mechanisms.
Page 9, lines 273–278, Discussion
Comment 4: Regarding the assessment of DNA damage (γH2AX), please clarify whether foci analysis was performed quantitatively and include the mean number of foci per cell, standard deviations, and the number of cells analyzed, rather than relying solely on representative images.
Response 4: We thank the reviewer for this valuable comment. In response, we quantified DNA damage using two complementary approaches: (i) γH2AX fluorescence intensity per nucleus was measured, background-corrected, and normalized to the 0 kBq control (=1), and (ii) the number of γH2AX foci per nucleus was quantified. The quantitative results, presented as dose–response bar graphs with mean ± SD, are now included in Supplementary Figure S3. A statistically significant increase in fluorescence intensity was observed in MP-CCS-SY cells at 100 kBq compared with 0 kBq (p = 0.03), whereas MG-63 cells showed a similar trend without reaching statistical significance. In addition, foci quantification demonstrated a dose-dependent increase in the mean number of γH2AX foci per cell. We have also added a description of these quantification procedures to the Methods section. The Results and Methods sections have been revised as follows:
Moreover, 211At-AAMT treatment induced DNA damage owing to the formation of DNA double-strand breaks (DSBs) (Figure 3), as evidenced by a significant increase in the total fluorescence intensity of γH2AX in the 100 kBq group relative to the 0 kBq group (Figure S3A), and by a dose-dependent increase in the mean number of γH2AX foci per cell (Figure S3B).
Page 5, lines 139–143, Results
For quantification, γH2AX fluorescence intensity per nucleus was measured using Fiji/ImageJ. Regions of interest (ROIs) were defined around individual nuclei based on DAPI staining, and the integrated density was calculated. Background correction was performed by subtracting the product of the ROI area and the mean background intensity. The corrected integrated density values were normalized to the 0 kBq control (=1). For foci analysis, γH2AX foci were manually counted within at least 50 nuclei per condition, and the average number of foci per nucleus was calculated.
Page 12, lines 402–408, Materials and Methods
Supplementary Figure S3
Comment 5: For the in vivo experiments, it is important to clearly state the number of animals used per group in each assay,
Response 5: We appreciate the reviewer’s valuable suggestion. The number of animals used per group has been clearly stated in the Results section and added to the legend of Figure 4 and 5.
Page 12, lines 410, 418, 420 and 428
Figure 4
Figure 5
Comment 6: and when presenting tumor growth curves, to specify the exact statistical tests applied (e.g., repeated measures ANOVA, Mann–Whitney test) and to report exact p-values for key comparisons, as sometimes only asterisks are shown.
Response 6: We thank the reviewer for highlighting this important point. We have clarified the statistical methods in both the Methods section and the figure legends. End-point tumor volumes were compared between groups using the Mann–Whitney U test, and exact p-values for the main comparisons have been added to the Results section as follows:
By contrast, a single administration of 211At-AAMT at a dose of 1 MBq significantly inhibited tumor growth (p = 0.008) without any noticeable reduction in the average body weight. Similarly, in 143B xenograft models, a single administration of 211At-AAMT resulted in remarkable tumor growth suppression, and consistent results were observed in terms of tumor weight (p = 0.001) (Figures 4B and S4).
Page 6, lines 164–168, Results
End-point tumor volumes were compared with the control using the Mann–Whitney U test; exact p-values are indicated in the Results.
Page 6, lines 178–179, Figure 4
Statistical comparisons for in vitro assays were performed using Student’s t-test, whereas differences in animal study data were evaluated using the Mann–Whitney U test. A two-tailed p-value of <0.05 was considered statistically significant.
Page 13, lines 452–455, Materials and Methods
Comment 7: It would also be valuable to briefly comment on whether any additional adverse events were observed apart from body weight changes, such as organ toxicity, and to indicate whether the follow-up period was sufficient to detect potential delayed effects.
Response 7: We thank the reviewer for this important comment and sincerely apologize that in the present study toxicity evaluation was limited to body weight monitoring, without systematic assessment of organ-specific toxicity, and that the observation period may not have been sufficient to detect delayed adverse events. To address this concern, we have revised the Discussion to cite previous studies and to clarify that while our current toxicity assessment was limited, prior reports have shown only transient and reversible changes without irreversible late effects and no sustained organ toxicity. We have revised the Discussion accordingly, and the revised text is as follows:
Although the adverse events associated with 211At-based alpha therapy in humans have not yet been comprehensively reported, previous toxicity studies of [211At]PSMA-5 demonstrated transient changes, with no irreversible toxicity observed 14 days after administration [18]. Consistently, our previous toxicity study of 211At-AAMT, with follow-up for 28 days, showed transient hematopoietic suppression that subsequently recovered, with no pathological abnormalities in major organs [16]. Notably, in the present study, although doxorubicin induced significant weight loss at effective doses, a single dose of 211At-AAMT suppressed tumor growth without causing weight loss.
Page 9, lines 260–267, Discussion
Accordingly, we emphasize in the revised Discussion the need for extended follow-up and organ toxicity evaluation in future studies, and we have incorporated this point into the limitation section as follows:
Fifth, comprehensive toxicity assessments, including off-target effects at LAT1 expression sites in normal tissues such as the blood–brain barrier and placenta, as well as long-term follow-up to detect potential delayed adverse events, are essential prior to clinical application.
Page 9, lines 283–286, Discussion
Comment 8: In the histopathology section, while the inclusion of representative images is appreciated, a quantitative analysis of the proportion of Ki-67- and γH2AX-positive cells, together with standard deviations and sample sizes, would reinforce the robustness of the findings.
Response 8: We thank the reviewer for this helpful comment. In response, we have quantitatively analyzed the proportion of Ki-67– and γH2AX–positive cells in both MP-CCS-SY and 143B xenografts and included the results in Figure 5 with standard deviations and sample sizes (Ki-67, n=5; γH2AX, n=4). We have also added the exact p-values for both tumor models to the Results section. The revised text now states:
The proportion of Ki-67-positive cells was significantly lower in the 211At-AAMT-treated group than in the control (MP-CCS-SY: p < 0.001; 143B: p = 0.001) and doxorubicin-treated groups (MP-CCS-SY: p = 0.003; 143B: p = 0.003), indicating a potent antiproliferative effect of 211At-AAMT (Figures 5A and 5B). Furthermore, histological analysis at 24 h post-treatment revealed a significantly higher proportion of γH2AX-positive cells in the 211At-AAMT group than in the doxorubicin group (MP-CCS-SY: p = 0.01; 143B: p < 0.001), suggesting that 211At-AAMT exerts its antitumor effect through DNA damage, consistent with in vitro findings.
Page 7, lines 183–194, Results, Figure 5
Comment 9: Throughout the results section, please standardize the reporting of numerical values and units, clarify statistical significance thresholds, and always indicate the sample size (n) in figure legends and tables.
Response 9: We thank the reviewer for this helpful comment. In response, we carefully revised the Results section and figure legends. Specifically, we have:
- Added the sample size (n) to all relevant Figure legends and Results descriptions.
- Standardized the concentration units by changing “nmol/L” to “nM” for consistency with other values.
- Clarified the statistical significance threshold, which is described in the Materials and Methods section as follows:
All results are presented as means ± standard deviation (SD). Statistical comparisons for in vitro assays were performed using Student’s t-test, whereas differences in animal study data were evaluated using the Mann–Whitney U test. A two-tailed p-value of <0.05 was considered statistically significant.
Page 13, lines 452–455, Materials and Methods
We believe these revisions have improved the clarity and consistency of data presentation throughout the manuscript.
Figure 1
Figure 2
Figure 4
Figure 5
Supplementary Figure S2
Supplementary Figure S3
Supplementary Figure S4
Supplementary Figure S5
Supplementary Table S1
Page 4, lines 122–126, Results
Page 13, lines 452–455, Materials and Methods
Comment 10: Finally, in interpreting the results, it would be helpful to explicitly acknowledge the limitations of the xenograft models, noting that results from immunodeficient mice may not fully translate to the human setting, and that the retention of LAT1 expression after 211At-AAMT treatment should be confirmed in more complex models or additional preclinical studies.
Response 10: We thank the reviewer for this important comment. We fully agree that the limitations of xenograft models should be explicitly acknowledged and that the observation of LAT1 expression maintenance requires further validation. Accordingly, we have revised the Discussion section to expand the limitations paragraph as follows:
Despite these promising results, this study has some limitations. First, the xenograft model used in this study did not fully recapitulate the complex tumor microenvironment and immune system of human sarcomas, and thus the findings, including the retention of LAT1 expression after 211At-AAMT treatment, may not be entirely extrapolated to clinical settings. Confirmation in more advanced preclinical systems, such as patient-derived xenografts, will be necessary.
Page 9, lines 268–273, Discussion

Round 2
Reviewer 1 Report
Comments and Suggestions for Authors
The authors have addressed the majority of the questions raised, and included new data in SI.